# Autism and ADHD in the Era of Big Data; An Overview of Digital Resources for Patient, Genetic and Clinical Trials Information

**DOI:** 10.3390/genes13091551

**Published:** 2022-08-28

**Authors:** Faris M. Abomelha, Hesham AlDhalaan, Mohammad Ghaziuddin, Nada A. Al-Tassan, Bashayer R. Al-Mubarak

**Affiliations:** 1King Salman Center for Disability Research, P.O. Box 94682, Riyadh 11614, Saudi Arabia; 2Center for Genomic Medicine, King Faisal Specialist Hospital and Research Center, P.O. Box 3354, Riyadh 11211, Saudi Arabia; 3Center for Autism Research, King Faisal Specialist Hospital and Research Center, P.O. Box 3354, Riyadh 11211, Saudi Arabia; 4Department of Psychiatry, University of Michigan, Ann Arbor, MI 48109, USA

**Keywords:** Autism, ADHD, database, patient registry, genetic variation database, clinical trials

## Abstract

Even in the era of information “prosperity” in the form of databases and registries that compile a wealth of data, information about ASD and ADHD remains scattered and disconnected. These data systems are powerful tools that can inform decision-making and policy creation, as well as advancing and disseminating knowledge. Here, we review three types of data systems (patient registries, clinical trial registries and genetic databases) that are concerned with ASD or ADHD and discuss their features, advantages and limitations. We noticed the lack of ethnic diversity in the data, as the majority of their content is curated from European and (to a lesser extent) Asian populations. Acutely aware of this knowledge gap, we introduce here the framework of the Neurodevelopmental Disorders Database (NDDB). This registry was designed to serve as a model for the national repository for collecting data from Saudi Arabia on neurodevelopmental disorders, particularly ASD and ADHD, across diverse domains.

## 1. Introduction

In the fields of health and biomedical research, the terms “patient registry” and “database” are mainly used to describe the collection and storage of datasets in an organized system that enables users to query, retrieve and sometimes extract information [1,2]. Currently, there is no consistent definition of the term “patient registry” or a set of defining characteristics in the literature. This makes the distinction between patient registries and simple databases or non-registry data repositories less clear. Regardless of terminology, both types of data systems can be sources of large amounts of valuable information for various stakeholders. Such data systems could be designed to serve each stakeholder’s specific interests and needs. Healthcare stakeholders include patients, healthcare providers (professionals and facilities), governmental parties, biopharmaceutical industries, regulatory agencies and researchers (academics and scientists).

For individuals with disabilities, be it intellectual, developmental or physical, registries have far-reaching benefits beyond optimizing healthcare. They could serve as a tool, assisting with achieving the best possible quality of life for those afflicted by these conditions. Information obtained from these registries can be leveraged to improve various areas of life, such as education, employment, as well as the social and physical environments.

Here, we describe the different types of patient registries, clinical trial registries and genetic databases concerned with ASD and ADHD. We also discuss their strengths and limitations. In our view, these two types of developmental disabilities deserve special attention for many reasons that could be boiled down to (1) the challenges associated with diagnosis and (2) the benefits of early intervention. Lack of definitive clinical, genetic or biochemical tests (except for rare monogenic forms) and the fact that these conditions exist along a spectrum, with subtle or overlapping features, render them harder to identify. Moreover, appropriate intervention, if introduced early in life, can significantly improve the quality of life of those affected and their families.

## 2. ASD Patient Data Systems

Because there is no clear distinction between the terms “database” and “registry”, we used the all-encompassing term “data system” throughout this section when referring to registries/databases. We initially aimed to review and summarize the key features of patient data systems dedicated to ASD and ADHD; however, we were only able to find information related to ASD, but none related to ADHD. Because trying to list all the existing data systems would be infeasible, we focused on identifying those described in the published literature and/or those with an official webpage. Our search was limited to articles and websites primarily available in English. Only active data systems are reviewed in this section (retired or obsolete data systems were deemed irrelevant, or those for which adequate information was not available were excluded). A total of nine data systems were identified; their key features, goals, strengths and limitations are summarized in Appendix A. Although all the covered data systems invariably served a similar purpose and collected similar variables, four different terms have been used in the literature/websites to identify them (database, registry, research registry and archive). While there is no formal definition that discriminates one term from the other, we noticed that the methods of data collection and patient recruitment were usually different (Appendix A).

All of the reviewed data systems have the same primary goal, that is advancing research by accumulating high quality data and linking researchers with potential participants and with one another. Under-representation of children from disadvantaged backgrounds or non-English speaking families is a major limitation that is common to most of these data systems.

The SFARI Base (https://www.sfari.org/resource/sfari-base/ (accessed on 15 August 2022)) [3] and the NIMH Data Archive (NDAR) (https://nda.nih.gov/about.html (accessed on 15 August 2022)) [4] are examples of premier big data resources, containing data derived from a large number of subjects, i.e., >250 k and >360 k, respectively. These figures are expected to rise due to the SFARI Base’s ongoing recruitment process and researchers depositing their new findings in NDAR. The SFARI Base is an online portal supported by Simons Foundation Autism Research Initiative that serves as a central database, housing various types of autism-related data, as well as banking biospecimens collected from the following four different cohorts: the Simons Simplex Collection (SSC), Simons Searchlight, Simons Foundation Powering Autism Research for Knowledge (SPARK) and Autism Inpatient Collection (AIC). The Interactive Autism Network (IAN), initially launched as a web-based portal for connecting researchers with individuals with ASD and their families, was established in 2006. This registry recruited over 60,000 participants before the project was closed in 2019 and transferred to SPARK [5].

While the SFARI Base contains data directly contributed by participants, NDAR, funded by the NIH, only holds data contributed by investigators and does not act as a repository for biospecimens. Although the SFARI Base and NDAR house an extensive set of data, they are generally limited in their geographical coverage (except for Simons Searchlight), being restricted to subjects residing in the USA or research projects funded by US-funding bodies (mainly the NIH). The International Collaboration for Autism Registry (iCARE), on the other hand, stands out as being the only one with multinational population coverage, spanning 6 countries, however, the limited access (data only available to member-sites) could be considered as a major downside [6].

Among the data systems reviewed herein, the Autism Genetic Resource Exchange (AGRE) is the longest standing. It was founded in 1997 as a collaborative venture by parents, physicians and researchers to accelerate the scientific progress of autism research. Currently, AGRE is supported by the Autism Speaks advocacy organization and funded by the National Institute of Mental Health [7]. Moreover, the MSSNG project houses data generated by whole genome sequencing of thousands of individuals from the AGRE and other cohorts [8]. This database was designed to create a resource of high quality “big data” that are available to researchers by leveraging cloud-based computing services. The latest data release summary (2019–2020) reports the accumulation of data from over 11 thousand individuals (including 5134 with ASD) that belong to families with either one or multiple affected members.

The Autism Treatment Network (ATN) Registry is another USA-based registry supported by Autism Speaks that was established to advance understanding and care delivery for children and adolescents with ASD and their families. The registry was curated to collect baseline measures at annual follow-up visits. The data can be queried by investigators to reach evidence-based recommendations or best practices in the management of ASD. Since its official launch in 2008, the registry cumulated over 7000 subjects from the USA and Canada [9]. 

As is the case with Autism Speaks, Autistica is a registered charity that was founded with the aim of improving the quality of life of individuals with ASD by facilitating and funding research. It is composed of a network of 50 UK child health teams and self-referral that has successfully recruited more than 13 thousand families [10].

Distinct from the other registers, the Autism Register is a continuation of a pre-existing paper-based register that was created in 1999 for the purpose of recording the incidence of ASD in the state of West Australia. In 2018, the register transitioned to an online portal that was designed to serve research by linking interested families with ongoing projects, identifying core mutual characteristics across ASD individuals and ultimately utilizing the outcome for better planning of essential health and educational services [9,11].

Readers can refer to Appendix A for a detailed comparison of all nine data systems.

## 3. ASD and ADHD Genetic Variation Databases

Although ASD and ADHD are two distinct disorders, they often co-exist or manifest as comorbidities in rare syndromes with recognized genetic causes, such as Fragile X and tuberous sclerosis. The genetic etiology of sporadic or non-syndromic forms of ASD and ADHD remains largely elusive [12,13]. The precise genetic and biological mechanisms that underpin the non-syndromic form of these disorders also remain largely elusive. To date, 100s of genomic regions (comprising of single or multiple genes) have been implicated in ASD or ADHD susceptibility. The exponential growth in genomic discoveries witnessed in the past decade or so have been facilitated by the availability of genome-wide association studies and next generation sequencing. This abundance of data can be put to optimal use through the process of curation and collation into accessible databases. In this section, we will review ASD-related genomic databases that are publicly available and currently active (not retired, obsolete or under construction). We compiled a non-exhaustive list of seven databases, including the Autism database, SFARI GENE, Autism Knowledge database and four more (see Appendix A). Of note, to date, there is only one genetic database dedicated to ADHD, which is named ADHDgene [14]; however, its content has not been updated since 2014 and more recently, it has been removed from the web.

### 3.1. Autism Database (AutDB)

The database was created in 2007 by MindSpec, a non-profit organization with a mission to accelerate research on neurodevelopmental disorders by using cutting-edge bioinformatics tools for ongoing cataloguing of genes and variants associated with ASD susceptibility [15]. Its content is built on data extracted from published clinical and scientific studies, manually curated by expert researchers. The database in its latest version (AutDB 2.0) comprises of five interactive modules, including the human gene, animal model, protein interaction (PIN), copy number variants (CNV) and gene scoring. The human gene module, the core component in which all other modules are integrated, represents an exhaustive up-to-date reference for all human genes with a documented link to ASD. The genes compiled within this module are classified according to the type of supporting genetic evidence into rAut (genes implicated in rare monogenic forms); sAut (genes implicated in syndromic forms; iAut (small risk-conferring candidate genes); and fAut (functional candidates biologically relevant to ASD).

### 3.2. SFARI GENE

This database is a project funded by the Simons Foundation Autism Research Initiative (SFARI) and developed by MindSpec (SFARIGenes news [16]). Launched as a licensed version of AutDB in 2008, it serves the same aims and follows a similar framework and portal design. In a similar way to AutDB, this database applies a system biology approach through the implementation of the following three interactive data modules: human gene, animal models and CNV. The data contained in each module are derived entirely from peer-reviewed literature. Although very similar in concept, SFARI GENE differs from AutDB in the following three main aspects: (1) the animal models module includes only those created in mice, (2) it does not include any analytical tools, such as PIN and (3) it uses a more comprehensive scoring criteria that evaluates all available evidence and not only what is collected from genetic studies.

### 3.3. Autism Knowledge Database (AutismKB)

This database was first launched in 2011 by the Wei group from the Center for Bioinformatics at Peking University, China [17]. The primary goal of this database was to build a knowledge portal that provides comprehensive reviews and analyses of published information on ASD genomics. As with previous databases, AutismKB is comprised of data primarily originating from peer-reviewed literature. As for gene functional annotation, the developers collect extensive information on molecular function, genomic variants, homologous genes, reported animal models and expression profile from secondary databases. Entries are also linked to three additional neurological disorders databases (AlzGene [18], SzGene [19] and PDGene (http://www.pdgene.org/, accessed on 15 August 2022)) to identify overlapping genes. The second version of the knowledgebase (AutismKB 2.0), released in 2018, integrated both the KOBAS enrichment analysis tool and variant pathogenicity prediction tools to achieve a more sophisticated ranking [20].

### 3.4. VariCarta

This database is the most recent addition, launched in 2019 as part of ASD sequencing studies by Pavlidis Lab (Michael Smith Laboratories at the University of British Columbia [21]). What distinguishes it from the databases above is that it specializes in cataloguing single nucleotide variants (SNVs) with extensive annotation. The database was developed with the intention to address a number of issues pertaining to data aggregation across cohorts and studies, such as methodological inconsistencies, subject overlap and variations in variant reporting formats.

### 3.5. Structural Variant Databases

The above-mentioned databases are concerned with listing submicroscopic genetic events (CNVs and SNVs) that occur only in the protein-coding regions of the human genome. Currently, only two databases serve as a catalogue of microscopic genetic events (structural variations existing at the chromosomal level), the Autism Chromosome Rearrangement Database (ACRD) [22] and the Autism Genetic Database (AGD) [23]. ACRD was one of the research initiatives supported by The Center for Applied Genomics at The Hospital for Sick Children in Canada. The database was curated in 2004 to house autism-related chromosomal abnormalities and cytogenetic break points. The last update of the database content was in 2014, listing 1695 breakpoints and 372 gain/loss events across almost all chromosomes. While the scope of ACRD and AGD overlap, AGD offers the advantage of incorporating four main groups of non-coding RNAs, including microRNAs, small nucleolar RNAs, Piwi-interacting RNAs and small interfering RNAs. At the time of writing this manuscript, the AGD website (http://wren.bcf.ku.edu/ (accessed on 12 January 2021)) was removed from the public domain (access attempted in February 2021) and the database was retired (personal communication with the author Prof Talebizadeh).

## 4. Clinical Trials on ASD/ADHD

Despite continuous efforts, no effective medication has been approved for the treatment of core ASD symptoms [24]. Current treatments are primarily aimed at alleviating ASD-associated symptoms (e.g., irritability, inattention and hyperactivity) and are mainly based on compounds repurposed from other conditions with similar symptoms. However, potential targets for drug discovery or development continue to emerge as a result of the explosion of genomics and system neuroscience findings [25,26]. On the other hand, there are at least five approved medications (including stimulants and non-stimulants), with widely demonstrated effectiveness, for core ADHD symptoms. While these medications are safe and effective for the majority of patients, a considerable proportion of patients have inadequate responses, poor tolerance, contraindication or experience side effects. These issues, in addition to long-term effectiveness, are the focus of nearly all drugs in development for ADHD [27,28]. For both disorders, non-pharmacological interventions, such as behavioral-based therapies and social skills training, are usually recommended, and in some cases are regarded as a standard addition to drug treatment. Only a few types of behavioral therapies (e.g., behavioral parent training and applied behavioral analysis) are supported by sufficient evidence; however, the efficacy of these therapies, either alone or in combination with other behavioral-based interventions and/or medical treatments, can vary between cases and may be influenced by the sequence at which the intervention was delivered or the age at which it was introduced [27,29].

In this section, we will provide an overview of the clinical trials for ASD and ADHD. Here, only clinical trials listed in public registries recognized by the International Committee of Medical Journal Editors (ICMJE) and accepted as primary registries in the WHO ICTRP Network were considered. Around 18 ICMJE-recognized clinical trial registries exist across different continents (Appendix A). The search for ADHD and ASD clinical trials was conducted by entering one of the following terms: “autism” or “autism spectrum disorder” or “attention deficit and hyper activity disorder” or, “attention deficit disorder with hyperactivity” or “ADHD” into the condition/disease search field of the registry website. The term/synonym that returned the highest number of search results was then selected for subsequent analysis. Registries that returned zero results, did not have the option to filter trials by condition/disease or displayed a technical error at the time of search were excluded from the analysis (see Appendix A for information).

Among the reviewed registries, ClinicalTrials.gov (accessed on 15 August 2022) houses the largest number of clinical studies on ASD and ADHD (Appendix A). This database is maintained by the National Library of Medicine located at the National Institutes of Health (NIH) and it contains information on clinical studies that span a wide range of medical conditions conducted not only in the USA but also in 220 different countries. This, perhaps, is not surprising for reasons including, but not limited to, the following: (1) the USA, despite the emergence of potential competitors, still maintains its position as a world leader in R&D, with a reported expenditure of more than half a trillion US dollars for 2018 alone, accounting for ~28% of the global spending [30]; (2) an annual budget of 30 billion US dollars is allocated from federal funds to the NIH, a large proportion of which (>80%) goes to support biomedical and health research [31]; (3) USA is the hub for the world’s largest pharmaceutical and biomedical companies.

The majority of the clinical trials listed within the registries analyzed herein adopted an interventional design strategy. Primary research is broadly categorized into interventional and observational studies. Interventional studies, also referred to as experimental studies, are those where the investigator(s) assign the participants to groups that either receive or do not receive a certain intervention to determine its effect on health-related outcomes. Contrary to the former, in observational studies, the investigator(s) does not interfere as part of the study design; instead, they merely observe and document the relationship between the factors and outcomes throughout the natural course of events. It is worth noting that patient registries can be considered a type of observational study as they can serve as grounds for generating and testing a hypothesis through their readily accessible wealth of data. Undoubtedly, each type of clinical study has its own advantages and limitations. For instance, conducting interventional studies may raise major ethical issues if addressing the research question requires subjecting participants to harm or withholding an effective and proven intervention deliberately. In such cases, observational studies can be the obvious option. However, they are often prone to confounding and bias and their results can be disputable.

Another common feature of the clinical trials reviewed herein is that they almost exclusively target children (school-aged and adolescents). The relative lack of studies conducted on adults means that there is little evidence to guide intervention in adults because as individuals transition from childhood to adulthood, the brain undergoes dynamic changes (due to compensatory mechanisms and gene environment interplay across the lifespan) that, in turn, can alter their needs and their response to intervention.

## 5. Neurodevelopmental Disorders Database (NDDB)

The overwhelming majority of information on individuals with ASD/ADHD that is made available through different types of data systems (genetic databases, patient registries and clinical trials registries) is largely curated from European (and to a lesser extent) Asian populations. Other populations/ethnic groups are severely understudied and, consequently, are underrepresented.

Our group is interested in studying neurodevelopmental disorders, particularly ASD and ADHD in the Saudi population. Among the six GCC countries, Saudi Arabia is the largest in terms of area and population size. However, no precise estimates exist and the available data are mainly anecdotal and lack nationwide coverage. In addition, government or non-government-maintained ASD or ADHD registries/databases are non-existent. Acutely aware of this significant gap in knowledge, we have created a demo database, the Neurodevelopmental Disorders Database (NDDB), with the intention of developing it into a public website that is accessible to the general public with restrictions. In this section, we provide an overview and description of the NDDB framework.

### 5.1. Purpose of NDDB

(1) To provide a searchable listing of individuals with neurodevelopmental disorders (primarily ASD and ADHD).

(2) To provide a central repository of searchable epidemiological, clinical, behavioural and genetic data.

### 5.2. Potential Data Contributors/Users

Data contributors or users are those that are anticipated to deposit data and may as well benefit from knowledge of the data and examples include hospitals, schools, rehabilitation centers, government bodies and researchers.

### 5.3. Design, Access and Privacy Protection

The database proposed in this paper is implemented using NoSQL, accessible via a Web API portal, that will be hosted and stored at King Salman Center for Disability Research (KSCDR) data center. Web-based applications allow the database to be compatible with most platforms; therefore, it accessible by a broader community. The database structure is illustrated in Figure 1 and is based on client–server architecture (Figure 2). Safeguards will be implemented to protect the collected health information in compliance with The Health Insurance Portability and Accountability Act of 1996 (HIPAA) privacy and security rules. The data, in both forms (in transit and at rest), will be non-identifiable and encrypted with the AES-256 algorithm.

## 6. Conclusions

Patient registries, genetic databases and clinical trial registries may have different primary aims; however, they all serve one ultimate purpose, that is, improving patient care and overall quality of life. Patient registries can help us to understand the natural history of a condition, identify potential research participants for clinical or preclinical studies and provide epidemiologic information. Genetic databases, on the other hand, can serve as a gateway for more in-depth understanding of disease mechanisms from mode of inheritance to revealing underlying biological pathways. The gained knowledge, therefore, holds promise for uncovering drug targets, developing diagnostic tests and application of precision medicine that can be evaluated through clinical trials.

In the world of big data, ADHD lags behind ASD; the reasons may not be readily apparent. Why does ADHD receive less attention compared to ASD? Is it because it can be more manageable or in some cases, symptoms can wane with age? These questions remain to be answered.

Finally, the fast-paced development and adoption of data sharing indicates that we are not far from harnessing the full potential of big data to achieve optimal health and well-being for all. An ideal system would be one that enables interoperability within and across organizational, regional and national boundaries, while safeguarding privacy and data security. Such a model can offer timely and seamless mobility of information across the complete spectrum of care and services.

## Figures and Tables

**Figure 1 genes-13-01551-f001:**
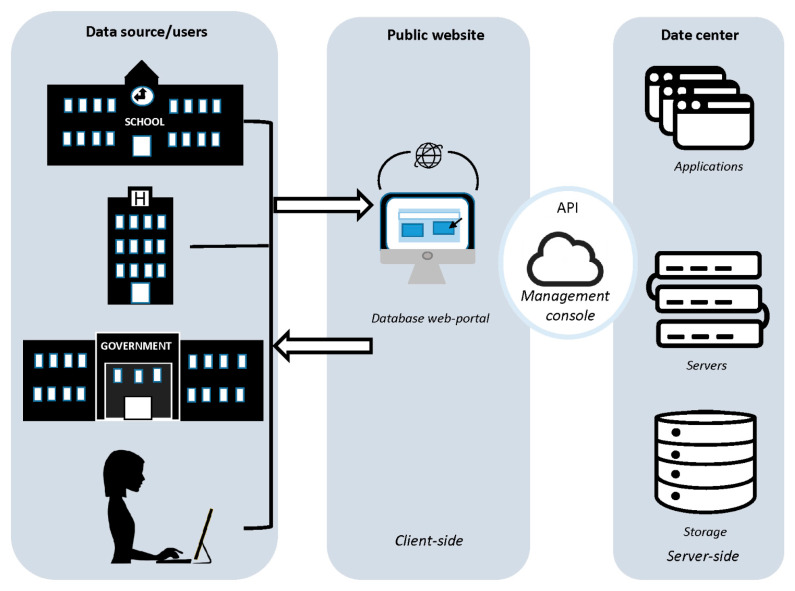
Schematic illustration of the database’s general structure and data flow. The database will include the following three modules: client-side, server-side, and management console. Users will input the data variables (listed in Appendix A) through a form-based interface that does not require special skills.

**Figure 2 genes-13-01551-f002:**
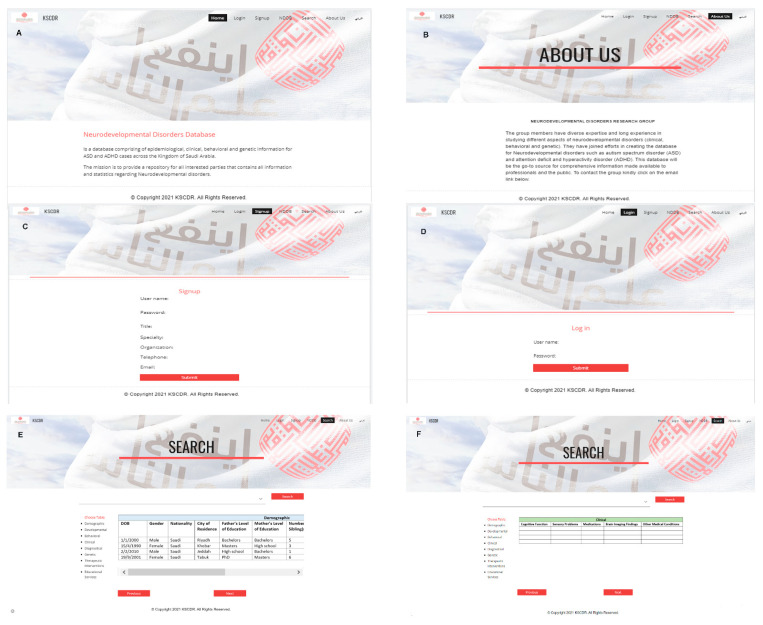
Screen shots of the user interface: (**A**) the homepage, (**B**) the About Us page, (**C**) signup page, (**D**) log in page; (**E**,**F**) search function. Users will be able to query data by keyword, data category or collaborator (contributing entity). User level privileges are shown in Appendix A.

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
