# Peer review of "Autism and ADHD in the Era of Big Data; An Overview of Digital Resources for Patient, Genetic and Clinical Trials Information"

_genes, 2022, doi:10.3390/genes13091551_

Round 1

Reviewer 1 Report

This is a review article about the interesting subject of big data (including data from patients' registries, clinical trials and genetic databases) related to autism and ADHD.

The introduction section describes in a comprehensive way the background of the current review.

The results are given in a detailed manner.

However, the structure of this manuscript needs extended improvement before publication. After the introduction section, which has number 1, there are no numbers used in headings or subheadings.

English language is generally ok, but there are some issues that need to be addressed before publication. For example, in line 95 the term "Founded in" should be replaced by "It was founded in".  In lines 103 and 183 a space should be placed before the parentheses. The sentence in lines 210-211 the sentence should be modified as follows: "The term/synonym, that returned highest number of search results, was then selected for subsequent analysis.

Finally, the conclusion section could perhaps be a bit more extended, describing in greater detail the differences between big data about ADHD and ASD and the possible reasons for these differences.

Author Response

Reviewer (1):

This is a review article about the interesting subject of big data (including data from patients' registries, clinical trials and genetic databases) related to autism and ADHD.

The introduction section describes in a comprehensive way the background of the current review.

The results are given in a detailed manner.

- We thank the reviewer for their positive note.

However, the structure of this manuscript needs extended improvement before publication. After the introduction section, which has number 1, there are no numbers used in headings or subheadings.

- We thank the reviewer for noticing this formatting detail. We have removed the numbering system and applied the following structure throughout; titles of main sections in short bold headings and titles of subsections in italic. 

English language is generally ok, but there are some issues that need to be addressed before publication. For example, in line 95 the term "Founded in" should be replaced by "It was founded in". 

- The sentence had been edited accordingly.

In lines 103 and 183 a space should be placed before the parentheses.

- The indicated lines have been screened for any typing issues and corrected accordingly.

The sentence in lines 210-211 the sentence should be modified as follows: "The term/synonym, that returned highest number of search results, was then selected for subsequent analysis.

- Punctuation corrections have been done as suggested.

Finally, the conclusion section could perhaps be a bit more extended, describing in greater detail the differences between big data about ADHD and ASD and the possible reasons for these differences.

- What we have noticed while preparing this review is the almost complete lack of publicly accessible data systems on ADHD as we have mentioned in the main text (lines 54-56)

“We initially aimed for reviewing and summarizing the key features of patient data systems dedicated to ASD and ADHD; however, we were only able to find information related to ASD but none related to ADHD”

(Lines 158-160) “Of note, to date there is only one genetic database dedicated to ADHD named ADHDgene [14]; however, its content has not been updated since 2014 and more recently had been removed from the web”

Our intention was to highlight this gap in knowledge and pose the question as to why ADHD receives less attention despite it being relatively more common than ASD. This question was put forward to the reader in the concluding remarks

Lines (347-349) “In the world of big data, ADHD lags behind ASD, the reasons may not be readily apparent.  Why does ADHD receive less attention compared to ASD?  Is it because it can be more manageable or in some cases, symptoms can wane with age? These questions remain to be answered.”

Therefore, conducting a comparison between available and accessible big data on ADHD vs ASD is currently not possible due to the limited information on ADHD.

Reviewer 2 Report

This manuscript is a useful addition to the literature in the context of review of current resources for ASD and ADHD. However, I have major concerns about the new Neurodevelopmental Disorders Database (NDDB).

I have a few points that need to be addressed:

1. ASD patient data systems section, "Due to space restrictions, only 4 data systems will be discussed in this section (readers can refer to Supplementary Table 1 for detailed comparison of all 8 data systems)." Please provide a rationale for selected these 4 from the 8 possible data systems. I don't really understand the space restrictions issue, why not just discuss all 8 data systems?

2. Neurodevelopmental Disorders Database (NDDB): "...we have created demo database, Neurodevelopmental Disorders Database (NDDB), with the intention of developing it into a public website accessible to the general public with restrictions." Please detail what restrictions will be in place and why?

3. Purpose of NDDB: "To provide a central repository of searchable epidemiological, clinical, behavioural and genetic data." Please provide detail on how this database will be curated and regulated. There is no mention of ethics approval for research - how will this be monitored? Will this database align with FAIR principles (https://www.go-fair.org/fair-principles/). How will this database align with the other databases discussed in the manuscript?

Author Response

Reviewer (2)

This manuscript is a useful addition to the literature in the context of review of current resources for ASD and ADHD. However, I have major concerns about the new Neurodevelopmental Disorders Database (NDDB).

 - We thank the reviewer for their positive note.

I have a few points that need to be addressed:

  1. ASD patient data systems section, "Due to space restrictions, only 4 data systems will be discussed in this section (readers can refer to Supplementary Table 1 for detailed comparison of all 8 data systems)." Please provide a rationale for selected these 4 from the 8 possible data systems. I don't really understand the space restrictions issue, why not just discuss all 8 data systems?

- We thank the reviewer for drawing our attention to this point. We were under the impression that a word limitation applies on review articles. This section now contains description and discussion of all 9 data systems.

  1. Neurodevelopmental Disorders Database (NDDB): "...we have created demo database, Neurodevelopmental Disorders Database (NDDB), with the intention of developing it into a public website accessible to the general public with restrictions." Please detail what restrictions will be in place and why?

- Details of user privileges and access restrictions are going to be decided at a later stage with all stake holders input as this is intended to be a national database. Usage, access and data privacy and security will be in compliance with international regulations (HIPAA) as well as national (Saudi) regulations issued by Saudi Data & AI authority

  1. Purpose of NDDB: "To provide a central repository of searchable epidemiological, clinical, behavioural and genetic data." Please provide detail on how this database will be curated and regulated.

- The database will be curated/populated with data/information from different contributors as mentioned in “Potential data contributors/users” section

- This database is intended to serve as demo for a national database, the role of regulation and governance will be undertaken by the ministry(s) that will act as primary stake holders.

- Usage, access and data privacy and security will be in compliance with international regulations (HIPAA) as well as national (Saudi) regulations issued by Saudi Data & AI authority.

There is no mention of ethics approval for research - how will this be monitored? Will this database align with FAIR principles (https://www.go-fair.org/fair-principles/). How will this database align with the other databases discussed in the manuscript?

There is no mention of ethics approval for research how will this be monitored?

- This database is intended to serve as demo for a national database, the role of regulation and governance will be undertaken by the ministry(s) that will act as primary stake holders.

- Data contributors (researchers) will be requested to provide proof of relevant IRB  approvals to a data access committee that will govern the database, if the contributor is non-research based i.e. hospital or school then the appropriate consents will be provided.

Will this database align with FAIR principles (https://www.go-fair.org/fair-principles/). How will this database align with the other databases discussed in the manuscript?

- The plan for the NDDB is to start as a national registry accessible to users within Saudi Arabia to fulfil nation wide needs.

- In the future and depending on the government policy and strategy, the database or parts of it may be made accessible to international users for which data democracy principles or guidelines such as FAIR may be taken into consideration.